# Risk factors and prognosis of acute lactation mastitis developing into a breast abscess: A retrospective longitudinal study in China

Daxue Li[1][◉], Jiazhen Li[2][◉], Yuan Yuan[3][‡], Jing Zhou[1][‡], Qian Xiao[1][‡], Ting Yang[1][‡], Yili Li[1][‡], Lili Jiang[1][‡], Han Gao[1]*

**1** Department of Breast and Thyroid Surgery, Chongqing Health Center for Women and Children (Women and Children's Hospital of Chongqing Medical University), Chongqing, China, **2** Department of Medical Ultrasonics, General Hospital of Chonggang, Chongqing, China, **3** Department of Medical Record, Chongqing Health Center for Women and Children (Women and Children's Hospital of Chongqing Medical University), Chongqing, China

◉ These authors contributed equally to this work.
‡ These authors also contributed equally to this work
* 15223323120@163.com

**Data Availability Statement:** All relevant data are within the manuscript and its Supporting Information files.

## Abstract

### Background

Breast abscess is developed on the basis of acute mastitis, which will cause damage to the physical and mental health of lactating women and is an important factor affecting the rate of breastfeeding. This study examined the risk factors for mastitis to develop into breast abscess, and analyzed the distribution of pathogenic bacteria, bacterial resistance, and treatment outcome.

### Methods

The medical records of 316 cases of mastitis and 219 cases of breast abscess were retrospectively collected. We analyzed the bacterial distribution of mastitis and breast abscess, and compared the differences of bacterial drug resistance. Univariate analysis and binary logistic regression were used to analyze the following aspects: age, primiparity or not, history of breast surgery, body temperature, puerperium or not, onset time, located in the nipple/areolar complexe area or not, history of massage by non-professionals, staphylococcus aureus/methicillin-resistant staphylococcus aureus (MRSA) infection or not, diabetes and white blood cell count.

### Results

Of the 535 patients, 203 (37.9%) were positive for staphylococcus aureus. There were 133 (65.5%) cases of methicillin-sensitive staphylococcus aureus (MSSA) and 70 (34.5%) cases of MRSA. Concerning bacterial drug resistance, a statistical analysis showed that MSSA had high resistance rate to penicillin (96.2%), ampicillin (91%), clindamycin (42.9%) and erythromycin (45.9%). MRSA had a high resistance rate to penicillin (100%), ampicillin (98.6%), oxacillin (95.7%), erythromycin (81.4%), clindamycin (80%), and amoxicillin

**Funding:** This research was supported by the Joint Medical Research Program of Chongqing Municipal Health Commission and Chongqing Science and Technology Bureau (2020FYYX135). The funders had no role in study design, data collection and analysis, decision to publish, or preparation of the manuscript.

**Competing interests:** The authors have declared that no competing interests exist.

(31.7%). Risk factors for progression of mastitis to breast abscess include a body temperature<38.5°C, a postpartum time $\geq$ 42 days, an onset time $\geq$ 2 days, lesions in the nipple/areolar complex area, a history of massage by non-medical staff and bacterial cultures for milk or pus that test positive for staphylococcus aureus or MRSA (P < 0.001).

## Conclusions

The most common pathogenic bacteria of mastitis and breast abscess is staphylococcus aureus. There are many risk factors for mastitis to develop into breast abscess. We should take effective measures for its risk factors and select sensitive antibiotics according to the results of bacterial culture to reduce the formation of breast abscess.

## Introduction

The importance of breastfeeding for maternal and child health has become an international consensus. Both the World Health Organization (WHO) and United Nations International Children's Emergency Fund (UNICEF) recommend exclusive breastfeeding for infants for the first six months of life [1]. The global rate of exclusive breastfeeding for infants aged 0–6 months is 43% [2], while the rate in China is lower than the world average. According to a 2019 report by the China Development Research Foundation, only 29.2% of infants aged 0–6 months are exclusively breastfed in China [3]. Mastitis or breast abscess during lactation is an important factor affecting breast-feeding rate [4,5]. One of the most common complications of mastitis or breast abscess is the cessation of breastfeeding [6]. Scott JA reports that about 10 percent of women with mastitis stop breastfeeding [7].

With an incidence rate of 1–33%, acute mastitis is a common postpartum disease in lactating women [8]. Nipple fissures and milk stasis often cause it during breastfeeding [9]. Due to a decline in their defense ability, bacteria through the milk ducts retrograde into the mammary gland, leading to infection. Patients often present with breast redness, swelling, tenderness, and poor milk discharge in the early stage. As the disease progresses, it may lead to the formation of a lump and be accompanied by fever, chills, fatigue, headache, and other symptoms. If the inflammation is not controlled in time, about 4.6–11% of patients eventually develop a breast abscess [10,11]. Without treatment, some patients may discharge pus through the skin, and ulcers may form. Many lactating women stop breastfeeding if an abscess causes a loss of milk or experience pain, or have to undergo treatment. To reduce the incidence of breast abscess by early prevention and intervention, we explored the risk factors associated with the development of a breast abscess due to breast mastitis.

## Materials and methods

### Study design

The retrospective longitudinal study was performed at Chongqing Health Center for Women and Children, which is a specialist general hospital for the treatment of women's and children's diseases. About 17,000 women give birth in our hospital every year. Data were enrolled for all patients diagnosed with mastitis or breast abscess between January 2019 and December 2020. The data were extracted from medical records and evaluated by two independent clinical physicians (Qian Xiao and Ting Yang). The anonymity of medical data was strictly monitored by a doctor (Yili Li), who had access to participants' information during and after data collection.

To protect patient privacy, the data is anonymized and the code does not include any information about the patient's identity. This study was approved by the Ethics Committee of Chongqing Health Center for Women and Children. Due to the retrospective, anonymized nature of the study, patient informed consent was waived.

### Identification of the study population

All adult patients that were diagnosed with lactation acute mastitis or breast abscess were screened for the study. Acute mastitis was diagnosed if any of the following criteria were met: (1) local redness of the breast, with or without a rise in skin temperature; (2) a systemic inflammatory reaction, such as chills, headache, and fatigue; (3) a body temperature > 37.3˚C; or routine blood test results that showed increased white blood cells (WBCs) or neutrophils or increased C-reactive protein levels. (4) Patients with positive milk culture. Diagnostic criteria of breast abscess: in addition to a diagnosis of mastitis, they met the following selection criteria: (1) had non-echo areas or low echo area as confirmed by an ultrasound examination, and flow observed after pressure; or (2) had pus that could be extracted by needle aspiration.

### Date collection

The clinical data of the patients were collected, including data on age, primiparity or not, postpartum time, onset time, a history of breast surgery, fever or not, location of lesions in the nipple/areolar complex area or not, a history of massage by non-professionals, diabetes, and the results of bacterial cultures of breast milk or pus. If bacteria are isolated from milk or pus cultures, the type of bacteria will be recorded. If it is Staphylococcus aureus, we divide it into MSSA and MRSA based on drug sensitivity. The resistance of each staphylococcus aureus to different antibiotics was recorded.

### Statistical method

SPSS 22.0 software was used for the statistical analysis. We performed a univariate analysis to examine the risk factors of breast absceess formation and the difference in antibiotic resistance between MSSA and MRSA. Independent sample t-tests analyzed the measurement data, and chi-square tests analyzed the counting data. The test level was $\alpha = 0.05$. Risk factors for breast abscess formation were further analyzed using a multivariate analysis, which was performed using binary logistic regression analysis.

### Results

In total, 535 patients with lactation mastitis or a breast abscess, who had been admitted to our hospital, were included in this study. Among the patients, 316 (59.1%) were allocated to the breast inflammation group, and 219 (40.9%) were allocated to the breast abscess group. Patients had a mean age of 29 years. 439 (82.1%) patients had undergone first-time labor. 22 (4.1%) patients had breast surgery previously. 268 (50.1%) patients had a body temperature $\geq 38.5$˚C. 319 (82.1%) patients presented in the puerperium period (in 42 days after delivery). 246 (46.0%) patients visited the doctor within 2 days of the onset of the illness. 244 (45.6%) patients had lesions in the nipple/areolar complex area. 131 (24.5%) patients had received a breast massage by non-professional personnel before onset. 226 (42.2%) patients developed pathogenic bacteria in their milk or pus. 7 (1.3%) patients had diabetes. 365 (68.2%) patients had elevated routine leukocyte counts.

**Table 1. Bacterial cultures were obtained from the breast inflammation group and the breast abscess group.**

|  | Inflammation Group (n = 316) | | Abscess Group (n = 219) | |
|---|---|---|---|---|
|  | **n** | **%** | **n** | **%** |
| • Staphylococcus aureus | 65 | 20.6 | 138 | 63 |
| MSSA | 53 | 16.8 | 80 | 36.5 |
| MRSA | 12 | 3.8 | 58 | 26.5 |
| • Other bacteria | 18 | 5.7 | 5 | 2.3 |
| Streptococcus agalactiae | 8 | 2.5 | 1 | 0.5 |
| Coagulase-negative staphylococcus | 4 | 1.3 | 0 | 0 |
| Staphylococcus epidermidis | 1 | 0.3 | 1 | 0.5 |
| Pseudomonas aeruginosa | 2 | 0.6 | 0 | 0 |
| Klebsiella pneumoniae | 1 | 0.3 | 1 | 0.5 |
| Streptococcus dysgalactiae | 0 | 0 | 1 | 0.5 |
| Streptococcus salivarius | 0 | 0 | 1 | 0.5 |
| Escherichia coli | 1 | 0.3 | 0 | 0 |
| Enterobacter aerogenes | 1 | 0.3 | 0 | 0 |

## Etiological distribution and prognosis (Table 1)

Bacterial culture tests were performed on breast milk or pus for all patients at admission. Concerning the bacterial cultures, 203 of 226 patients tested positive for Staphylococcus aureus, 133 patients tested positive for MSSA, 70 patients tested positive for MRSA, and 23 patients tested positive for another type of bacteria (9 for Streptococcus agalactiae, 4 for Coagulase-negative Staphylococcus, 2 for Staphylococcus epidermidis, 2 for Pseudomonas aeruginosa, 2 for Klebsiella pneumoniae, 1 for Streptococcus galactiae subspecies, 1 for Streptococcus salivarius, 1 for Escherichia coli, and 1 for Enterobacter aerogenes).

Concerning the anti-infection therapy, Patients who did not have a history of a penicillin allergy were treated with an intravenous flucloxacillin injection. If a patient did not respond well to the treatment, the antibiotics were adjusted according to the bacterial culture results of the milk or pus. Patients' length of stay in the mastitis group ranged from 3–10 days (mean: 4.22 days). Due to poor therapeutic effects, the antibiotics had to be changed or adjusted for 30 patients. In 2 cases, the antibiotics were changed due to a drug-induced rash. Breastfeeding was discontinued in 2 cases due to mastitis.

Patients in the breast abscess group underwent a daily breast ultrasound. If pus was found, ultrasound-guided needle aspiration was used to remove the pus. In the breast abscess group, the length of stay ranged from 3–12 days (mean: 6.58 days), the average size of the abscess cavity was 4.5 cm, 211 patients received an average of 3.9 ultrasound-guided aspirations, and 8 patients underwent a small incision and drainage. Due to the poor therapeutic effects, the antibiotics had to be changed or adjusted for 13 patients. In 1 case, the antibiotic treatment was discontinued due to a drug-induced gastrointestinal reaction. Breastfeeding was discontinued in 12 cases due to the breast abscess.

## Resistance of Staphylococcus aureus to antibiotics (Table 2)

In this study, the bacterial cultures of the milk or pus of 203 of the 535 patients (37.9%) tested positive for Staphylococcus aureus. As stated above, 133 patients tested positive for MSSA and 70 for MRSA. Concerning bacterial drug resistance, a statistical analysis showed that MSSA had high resistance rate to penicillin (96.2%), ampicillin (91%), clindamycin (42.9%) and erythromycin (45.9%). MRSA had a high resistance rate to penicillin (100%), ampicillin

**Table 2. Drug resistance of Staphylococcus aureus.**

| Drugs | MSSA (n = 133) | | MRSA (n = 70) | | $x^2$ | P |
|---|---|---|---|---|---|---|
| | n | % | n | % | | |
| Penicillin | 128 | 96.2 | 70 | 100.0 | 3.842 | 0.072* |
| Ampicillin | 121 | 91.0 | 69 | 98.6 | 3.327 | 0.072 |
| Oxacillin | 2 | 1.5 | 67 | 95.7 | 177.248 | 0.000 |
| Amoxicillin | 1 | 0.8 | 26 | 37.1 | 49.561 | 0.000 |
| Clindamycin | 57 | 42.9 | 56 | 80.0 | 25.637 | 0.000 |
| Gentamicin | 10 | 7.5 | 2 | 2.9 | 1.052 | 0.305 |
| Erythromycin | 61 | 45.9 | 57 | 81.4 | 23.832 | 0.000 |
| Rifampin | 0 | 0.0 | 1 | 1.4 | 1.909 | 0.345* |
| Trimesulf | 7 | 5.3 | 2 | 2.9 | 0.187 | 0.665 |
| Tetracycline | 8 | 6.0 | 8 | 11.4 | 1.851 | 0.174 |
| Tobramycin | 12 | 9.0 | 0 | 0.0 | 6.713 | 0.009* |
| Trimethoprim | 12 | 9.0 | 5 | 7.1 | 0.211 | 0.646 |
| Teicoplanin | 3 | 2.3 | 0 | 0.0 | 1.603 | 0.553* |
| Vancomycin | 2 | 1.5 | 0 | 0.0 | 1.063 | 0.546* |
| Levofloxacin | 1 | 0.8 | 4 | 5.7 | 2.862 | 0.091 |

* Fisher's exact test.

(98.6%), oxacillin (95.7%), erythromycin (81.4%), clindamycin (80%), and amoxicillin (31.7%). The drug resistance rate of MRSA to ampicillin, oxacillin, amoxicillin, clindamycin, erythromycin and chloramphenicol was significantly higher than that of MSSA, the difference was statistically significant (P < 0.05).

### Univariate analysis of the progression of acute mastitis to breast abscess (Table 3)

The univariate analysis results showed that a body temperature<38.5˚C, a postpartum time ≥ 42 days, an onset time ≥ 2 days, lesions in the nipple/areola area, a history of massage by non-professionals, bacteria from milk or pus were cultured to staphylococcus aureus, and bacteria from milk or pus were cultured to MRSA, and an WBC count (p<0.001) were risk factors of abscess formation. Age, primiparity, a history of breast surgery, and diabetes were not significantly associated with abscess formation.

### Multivariate analysis of progression from acute mastitis to a mammary abscess (Table 4)

Multivariate analysis showed that a body temperature<38.5˚C, postpartum time ≥ 42 days, onset time ≥2 days, lesions location in the nipple/areola complex, a history of massage by non-professionals, and the bacterial culture of milk or pus was MRSA were independent risk factors for breast abscess formation (P < 0.001).

### Discussion

Lactation mastitis is an inflammatory reaction of the breast gland caused by milk stasis. If it is not treated properly, an abscess can form in a short time. Infected bacteria are mostly caused by staphylococcus aureus or streptococcus infections from the nipple but may also be caused by direct bacterial invasion [12,13]. In the present study, univariate and multivariate analyses

**Table 3. Univariate analysis of the progression of acute mastitis to breast abscess.**

| Risk Factor | Inflammation Group (n = 316) | | Abscess Group (n = 219) | | $x^2$ | $P$ |
|---|---|---|---|---|---|---|
| | n | % | n | % | | |
| Age | | | | | | |
| < 30 | 190 | 60.1 | 132 | 60.3 | 0.001 | 0.973 |
| ≥ 30 | 126 | 29.9 | 87 | 39.7 | | |
| Primiparity | | | | | | |
| Yes | 253 | 80.1 | 186 | 84.9 | 2.082 | 0.149 |
| No | 63 | 19.9 | 33 | 15.1 | | |
| History of breast surgery | | | | | | |
| Yes | 11 | 3.5 | 11 | 5 | 0.780 | 0.377 |
| No | 305 | 96.5 | 208 | 95 | | |
| Body temperature (°C) | | | | | | |
| < 38.5 | 98 | 31 | 169 | 77.2 | 110.230 | 0.000 |
| ≥ 38.5 | 218 | 69 | 50 | 22.8 | | |
| Puerperium (in 42 days) | | | | | | |
| Yes | 218 | 69 | 101 | 46.1 | 28.101 | 0.000 |
| No | 98 | 31 | 118 | 53.9 | | |
| Onset time (day) | | | | | | |
| < 2 | 233 | 73.7 | 13 | 5.9 | 239.38 | 0.000 |
| ≥ 2 | 83 | 26.3 | 206 | 94.1 | | |
| Located in the nipple/areolar complex area | | | | | | |
| Yes | 97 | 30.7 | 147 | 67.1 | 69.191 | 0.000 |
| No | 219 | 69.3 | 72 | 32.9 | | |
| History of massage by non-professionals | | | | | | |
| Yes | 26 | 8.2 | 105 | 47.9 | 110.355 | 0.000 |
| No | 290 | 91.8 | 114 | 52.1 | | |
| Staphylococcus aureus | | | | | | |
| Yes | 251 | 79.4 | 81 | 20.6 | 98.966 | 0.000 |
| No | 65 | 37 | 138 | 63 | | |
| MRSA | | | | | | |
| Yes | 12 | 3.8 | 58 | 26.5 | 10.863 | 0.001 |
| No | 304 | 96.2 | 161 | 73.5 | | |
| Diabetes | | | | | | |
| Yes | 3 | 0.9 | 4 | 1.8 | 0.241 | 0.623 |
| No | 313 | 99.1 | 215 | 98.2 | | |
| White blood cell count (×109/L) | | | | | | |
| < 9.5 | 75 | 23.7 | 95 | 43.4 | 26.689 | 0.000 |
| 9.5–14.9 | 154 | 48.7 | 95 | 43.4 | | |
| 15–19.9 | 69 | 21.8 | 23 | 10.5 | | |
| ≥ 20 | 18 | 5.7 | 6 | 2.7 | | |

showed that a risk factor for the occurrence of a breast abscess was bacterial cultures of milk or pus that were positive for Staphylococcus aureus. Studies have shown that Staphylococcus aureus is the most common pathogen in breast abscesses [14]. Moazzez et al. showed that Staphylococci were present in cultures in 50% of cases, and MRSA was present in 19% of community-acquired breast abscess isolates [15]. The detection rate of MRSA in our breast abscess

**Table 4. Multivariate analysis of progression from acute mastitis to mammary abscess.**

| Risk factor | B | SE | Wald | P | OR | 95%CI |
|---|---|---|---|---|---|---|
| Body temperature ≥ 38.5˚C | 1.173 | 0.540 | 4.719 | 0.030 | 3.232 | 1.122~9.313 |
| Postpartum time ≥ 42 day | 1.338 | 0.584 | 5.25 | 0.022 | 3.812 | 1.214~11.976 |
| Onset time ≥ 2 days | 3.601 | 0.586 | 37.804 | 0.000 | 36.647 | 11.627~115.508 |
| Lesions in nipple/areola area | 1.758 | 0.572 | 9.435 | 0.002 | 5.802 | 1.890~17.817 |
| History of massage by non-professionals | 2.589 | 0.925 | 7.842 | 0.005 | 13.319 | 2.175~81.562 |
| MRSA | 1.263 | 0.610 | 4.279 | 0.039 | 3.534 | 1.069~11.691 |
| White blood cell count (×109/L)* | -0.092 | 0.343 | 0.071 | 0.789 | 0.913 | 0.466~1.786 |
| **constant** | **-10.930** | **1.975** | **30.630** | **0.000** | **0.000** | |

* Reference group for variable is the group of white blood cell count<9.5×109/L.

group was 26.5%, which suggests that Staphylococcus aureus plays an important role in the development of breast abscesses.

In the stage of acute inflammation, the early use of antibiotics can achieve better efficacy. Our bacterial culture results revealed that MSSA had a high rate of drug resistance to penicillin (96.2%) and ampicillin (91%). MRSA had high resistance rates to a penicillin (100%), ampicillin (98.6%), oxacillin (95.7%), erythromycin (81.4%), and clindamycin (80%); thus, these should not be used as first choices in selecting antibiotics. Both groups were sensitive to gentamicin, rifampicin, cotrimoxazole, tobramycin, trimethoprim, chloramphenicol, and levofloxacin (≥ 92.9%). No drug resistance was found for teicolanin, vancomycin, linezolid, or quinuptin.

In our center, if a patient had no history of a penicillin allergy, flucloxacillin was administered intravenously. Only 43 patients (8%) changed antibiotics during treatment due to poor efficacy; thus, flucloxacillin had good effects. For patients who test positive for MRSA, the antibiotics need to be adjusted according to the clinical effects and the culture results. There is no need to change the treatment plans of patients who have been treated with non-sensitive antibiotics but show good clinical efficacy. One of the reasons may be the difference between in vitro test and in vivo efficacy and the other is that the theory of local treatment is another major factor to ensure the efficacy [16]. Young showed that up to 30% of MRSA soft tissue infections recover uneventfully after surgical drainage even when treated with antibiotics that were found to be insensitive based on culture sensitivity results [17]. Some studies have shown that in some patients with breast abscesses, drainage alone without antibiotics can achieve good results [18,19]. Ulitzsch et al. also suggested that Staphylococcus aureus, which is usually produced by β-lactamase, should be used with penicillinase-resistant antibiotics, such as flucloxacillin [20]. For patients with a β-lactamase allergy or for those who respond poorly, quinolone antibiotics should be used, and breastfeeding should be suspended. The Chinese Guidelines for the diagnosis and treatment of lactation mastitis also recommend the use of enzyme-resistant penicillins (e.g., benzacillin sodium), cephalosporin I (e.g., cefradin) or cephalosporin II (e.g., cefmetazole) for anti-infective therapy until the results of drug sensitivity tests are obtained [21]. The German Scientific Medical Association's guidelines also recommend that first and second generation cephalosporins or penicillins with beta-lactamase-inhibitor combinations which are safe for both mother and infant have become the antibiotic of choice [22].

Our study showed that the risk of breast abscess formation increased significantly if the onset of the disease was more than 2 days. As a lactation breast gland abscess is a bacterial infection, there is a positive correlation between the degree of infection and time. The long

duration of the disease and the prolonged duration of inflammation can also indicate a more severe infection, which increases the risk of abscess formation. For patients who have had the disease for > 2 days, attention should be paid to the physical and breast ultrasound examination results to ensure the timely detection of breast abscesses, especially deep breast abscesses, and avoid omission.

In our study, the univariate analyses suggested that body temperature lower than 38.5°C and a routine WBC count lower than 9.5×109/L were independent influencing factors related to abscesses. It may be that the pus and inflammation were confined to the mammary gland in some patients, and their systemic inflammatory reactions were not serious; thus, the temperature detection and WBC count were lower than those in the mastitis group. However, multivariate analysis suggested that white blood cell count was not a risk factor for abscess formation, and we considered that the possible reason for its non-statistical significance was the co-interference of other factors. Clinically, we tend to find that patients with abscesses are either in the acute phase, with marked redness and swelling and often high white blood cell counts, or in the stable phase, with limited pus and often normal white blood cell counts.

Our study showed that breast abscesses occur more frequently during the puerperium period (postpartum time in 42 days), which was associated with the mother's lack of breast-feeding experience. The incidence of breast abscesses was higher in the central area of the nipple than in the peripheral region. Inflammation in the nipple and areola region is more likely to obstruct the main milk duct, making it difficult to discharge milk and more likely for breast abscesses to form. Due to differences in customs, when breastfeeding is not smooth, Chinese women often turn to " non-professional massage therapist " or "old women with breastfeeding experience" for breast massage. Patients with a history of non-professional massage are more likely to cause damage to the breast ducts due to violent massages. This injury is also often located in the area of the nipple and areola, which also leads to the formation of breast abscesses. A meta-analysis from China also found that non-professional massage history was a risk factor for mastitis [23].

After the diagnosis of a breast abscess, open surgical drainage is traumatic, and patients experience pain when dressings are changed, which often leads to the discontinuation of breastfeeding. Ugly scars often form after incision and drainage, which can cause great harm to female patients both physically and mentally. Our patients with abscesses received ultrasound examinations every day during their hospitalization. If there was a no-echo area or liquid dark area, ultrasound-guided puncture and aspiration treatment were performed. Our data showed that the average length of hospital stay for breast abscess patients was 6.58 days, the average size of the abscess cavity was 4.5 cm, and the average number of puncture times was 3.9. Only 8 patients (3.7%) underwent small incision and drainage due to poor puncture effects, and only 10 patients (4.6%) stopped breastfeeding due to an abscess. Luo et al. also showed that after ultrasound-guided puncture treatment for patients with a postpartum breast abscess, the cure rate was 83.3% [24]. Other studies have shown that ultrasound-guided aspiration is an effective treatment for breast abscesses and should be recommended as a first-line treatment worldwide [10,25–27].

The results of this study may provide evidence-based information for the risk factors of mammary abscess during lactation in China, and help provide appropriate management advice, scientific prevention and treatment strategies and effective individualized care for the multidisciplinary team or related personnel involved in maternal and infant feeding management. However, there are some limitations to our study. First, differences in inter-study heterogeneity may affect the validity of statistical analysis due to potential confounding factors, such as sample size, design differences, and potential population characteristics. Secondly, our hospital is a specialized hospital for women and children, and all the cases included were

inpatients. Out-patients with mild symptoms were not included in this study, which may lead to selection bias. Finally, the study included only Chinese women, mostly primiparas, which may limit the generality and interpretation of the findings. However, our findings provide a risk factor for mastitis to develop into breast abscess, provide a reference for the prevention of abscess, and point to areas that need to be studied in the future.

## Conclusion

In conclusion, a body temperature<38.5˚C, a postpartum time $\geq$ 42 days, an onset time $\geq$ 2 days, lesions in the nipple/areola area, a history of massage by non-professionals and bacterial cultures for milk or pus that test positive for Staphylococcus aureus or MRSA are risk factors for the occurrence of a breast abscess. These findings have certain reference value for the prevention, treatment and individual nursing of breast abscess. In particular, the incidence of breast abscesses can be reduced by controlling modifiable risk factors.

## Supporting information

**S1 Data.**
(XLSX)

**S2 Data.**
(XLSX)

**S3 Data.**
(XLSX)

## Author Contributions

**Conceptualization:** Jiazhen Li.

**Data curation:** Qian Xiao, Ting Yang, Yili Li.

**Formal analysis:** Yuan Yuan, Jing Zhou.

**Funding acquisition:** Lili Jiang.

**Methodology:** Han Gao.

**Software:** Yuan Yuan, Jing Zhou.

**Writing – original draft:** Daxue Li, Jiazhen Li.

**Writing – review & editing:** Lili Jiang, Han Gao.

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
