## [Decision Letter · Decision Letter 0]

10 Mar 2022

PONE-D-21-29131

Risk factors and prognosis of acute lactation mastitis developing into a breast abscess

PLOS ONE

Dear Dr. Gao,

Thank you for submitting your manuscript to PLOS ONE. After careful consideration, we feel that it has merit but does not fully meet PLOS ONE’s publication criteria as it currently stands. Therefore, we invite you to submit a revised version of the manuscript that addresses the points raised during the review process.

Specifically: 

Key elements of study design  

The setting and location  

Any efforts to address potential sources of bias

Limitations of the study  

The innovation of the work

The diagnostic criteria of breast abscess

Presentation of the data

We look forward to receiving your revised manuscript.

Kind regards,

Forough Mortazavi

Academic Editor

PLOS ONE

A clean copy of the edited manuscript (uploaded as the new *manuscript* file).

“This research was supportted by the Joint Medical Research Program of Chongqing Municipal Health Commission and Chongqing Science and Technology Bureau（2020FYYX135）. The funders had no role in study design, data collection and analysis, decision to publish, or preparation of the manuscript.”

“Lili Jiang received the award. This research was supportted by the Joint Medical Research Program of Chongqing Municipal Health Commission and Chongqing Science and Technology Bureau（2020FYYX135）. URL：http://wsjkw.cq.gov.cn/.

Additional Editor Comments:

Dear authors,

Thank you for submitting the manuscript to PLOS ONE. According to the reviewers’ comments and my evaluation, the manuscript need careful attention and must be improved according to all the comments.

The Title of the study must indicate the study’s design with a commonly used term. The present title is suitable for a longitudinal study.

Introduction needs explanations for the necessity of the work. The rates of breastfeeding initiation and continuation in the country should be included in the introduction section.

PLS present key elements of study design in the methods section

PLS describe the setting and location including the number of deliveries and the type of hospital in which the data collected.

PLS describe any efforts to address potential sources of bias

PLS describe limitations of the study both in the abstract and in discussion.

Reviewers' comments:

Reviewer's Responses to Questions

**Comments to the Author**

1. Is the manuscript technically sound, and do the data support the conclusions?

Reviewer #1: No

Reviewer #2: Yes

2. Has the statistical analysis been performed appropriately and rigorously? 

Reviewer #1: Yes

Reviewer #2: Yes

3. Have the authors made all data underlying the findings in their manuscript fully available?

Reviewer #1: No

Reviewer #2: Yes

4. Is the manuscript presented in an intelligible fashion and written in standard English?

Reviewer #1: No

Reviewer #2: Yes

5. Review Comments to the Author

Reviewer #1: Thank you for your manuscript describing risk factors for mastitis to become a breast abscess. I have a few comments and queries:

1. Abstract- I don't think that your results as described lead to the conclusion that USS guided biopsy is essential - why not milk culture which is much less invasive?

2. Background - your background is important, I would emphasise the % of women who give up breastfeeding because of mastitis

3. Methods - How was the data collected? From hospital information systems or via hospital IDs? If so, the data was not anonymised directly. What was your denominator of deliveries, what type of hospital is the data collected from (high risk, normal etc.)

4. Results - your data summary of risk factors with % would be better presented in a table. Also, you do not mention in your abstract that the majority of women are primiparous - which could mean that better breastfeeding advice is needed to avoid breast abscess. Similarly bacterial cultures would be better in a table.Did you test whether MSSA or MRSA had higher risk of abscess formation? You dont mention in your abstract that the majority of women would have received inadequate treatment because of a high rate of penicillin resistance. Would this not mean that guidelines for therapy need to be changed?

5. Discussion - your discussion doesn't bring out the limitations to your study (these are in your conclusions but do'nt account for some of the factors above).

Reviewer #2: 1. This study explored the risk factors associated with the development of a breast abscess due to breast mastitis and clarified that the formation of breast abscesses is multifactorial. The data is very detailed and well represented.

2. But the author mentioned little about the innovation of this work, which makes the study more like a mere validation of previous ones. It may be helpful to rewrite the introduction and Discussion sections.

3.The diagnostic criteria of breast abscess are not very clear. Should non-echo areas and low echo areas be present simultaneously?

6. PLOS authors have the option to publish the peer review history of their article (what does this mean?). If published, this will include your full peer review and any attached files.

Reviewer #1: No

Reviewer #2: No

---

## [Author Response · Author response to Decision Letter 0]

23 May 2022

I have uploaded a file named "Respond to Reviewers". If you have any questions, please contact us.

---

## [Decision Letter · Decision Letter 1]

13 Jun 2022

PONE-D-21-29131R1Risk factors and prognosis of acute lactation mastitis developing into a breast abscess：a retrospective longitudinal study in China.PLOS ONE

Dear Dr. Gao,

Thank you for submitting your manuscript to PLOS ONE. After careful consideration, we feel that it has merit but does not fully meet PLOS ONE’s publication criteria as it currently stands. Therefore, we invite you to submit a revised version of the manuscript that addresses the points raised during the review process.

We look forward to receiving your revised manuscript.

Kind regards,

Forough Mortazavi

Academic Editor

PLOS ONE

Journal Requirements:

Additional Editor Comments (if provided):

Dear authors,

Thank you for resubmitting the manuscript together with some revisions. In order to move forward with this, however, we are requesting further revisions, particularly related to the following points:

1. There is a discrepancy between the authors’ response to PLOS ONE Clinical Studies Checklist regarding the ethics approval of the study and their remarks on the same subject in the methods section of the manuscript. In the PLOS ONE Clinical Studies Checklist, the authors state, “We did not obtain ethical approval, because our study was a retrospective study of medical records, and all data were fully anonymized.” But in the methods section, the authors state, “This study was approved by the Ethics Committee of Women and Children’s Hospital of Chongqing Medical University.”

2. With regard to the financial disclosure, the authors state, “This research was supported by the Joint Medical Research Program of Chongqing Municipal Health Commission and Chongqing Science and Technology Bureau（2020FYYX135. URL：http://wsjkw.cq.gov.cn/.” But the URL given by them does not work.

3. Also, the data supporting the contents of tables 1 and 2 are missing in the Excel file.

Reviewers' comments:

Reviewer's Responses to Questions

**Comments to the Author**

1. If the authors have adequately addressed your comments raised in a previous round of review and you feel that this manuscript is now acceptable for publication, you may indicate that here to bypass the “Comments to the Author” section, enter your conflict of interest statement in the “Confidential to Editor” section, and submit your "Accept" recommendation.

Reviewer #2: All comments have been addressed

2. Is the manuscript technically sound, and do the data support the conclusions?

Reviewer #2: Yes

3. Has the statistical analysis been performed appropriately and rigorously? 

Reviewer #2: Yes

4. Have the authors made all data underlying the findings in their manuscript fully available?

Reviewer #2: Yes

5. Is the manuscript presented in an intelligible fashion and written in standard English?

Reviewer #2: Yes

6. Review Comments to the Author

Reviewer #2: (No Response)

7. PLOS authors have the option to publish the peer review history of their article (what does this mean?). If published, this will include your full peer review and any attached files.

Reviewer #2: No

---

## [Author Response · Author response to Decision Letter 1]

28 Jul 2022

1. There is a discrepancy between the authors’ response to PLOS ONE Clinical Studies Checklist regarding the ethics approval of the study and their remarks on the same subject in the methods section of the manuscript. In the PLOS ONE Clinical Studies Checklist, the authors state, “We did not obtain ethical approval, because our study was a retrospective study of medical records, and all data were fully anonymized.” But in the methods section, the authors state, “This study was approved by the Ethics Committee of Women and Children’s Hospital of Chongqing Medical University.”

At the initial stage of submission, the ethical requirements of PLOS ONE mentioned that if the article is a retrospective analysis of medical data, it does not require the approval of the ethics committee. Therefore, we did not submit relevant documents to the ethics committee at the beginning. But after submitting the manuscript, one of the reviewers reminded us how we collected the data, and if it was through the hospital's information system, the data was not directly anonymous. We respect the opinions of the reviewers. Therefore, we supplemented the medical ethics approval document.

2. With regard to the financial disclosure, the authors state, “This research was supported by the Joint Medical Research Program of Chongqing Municipal Health Commission and Chongqing Science and Technology Bureau（2020FYYX135. URL：http://wsjkw.cq.gov.cn/.” But the URL given by them does not work.

This research was supported by the Joint Medical Research Program of Chongqing Municipal Health Commission and Chongqing Science and Technology Bureau（2020FYYX135）. The funded project was published on the official website of Chongqing Municipal Health Commission. We updated the URL, but the relevant document is in Chinese. Our approval is posted on page 9, line 4, serial number 135.URL http://wsjkw.cq.gov.cn/zwgk_242/wsjklymsxx/ylws_266434/yzgl_266435/gzxx/202009/W020200917627904433036.pdf. 

3. Also, the data supporting the contents of tables 1 and 2 are missing in the Excel file.

We uploaded the data for Tables 1 and 2.

---

## [Editor Report · Decision Letter 2]

1 Aug 2022

PONE-D-21-29131R2Risk factors and prognosis of acute lactation mastitis developing into a breast abscess：a retrospective longitudinal study in China.PLOS ONE

Dear Dr. Gao,

Thank you for submitting your manuscript to PLOS ONE. After careful consideration, we feel that it has merit but does not fully meet PLOS ONE’s publication criteria as it currently stands. Therefore, we invite you to submit a revised version of the manuscript that addresses the points raised during the review process.

We look forward to receiving your revised manuscript.

Kind regards,

Forough Mortazavi

Academic Editor

PLOS ONE

Journal Requirements:

Additional Editor Comments (if provided):

Dear authors,

Thank you for submitting the revised manuscript. In order to move forward with this, however, we are requesting further revisions including:

1. In table 4, the reference group for the ‘white blood cell count’ variable should be identified.

2. Lines 83-85 should be revised.: [(3) a body temperature > 37.3℃; or (4) routine blood test results that showed increased white blood cells (WBCs) or neutrophils or increased C reactive protein levels. (4) Patients with positive milk culture.] PLS check the text again.

3. I noticed that in your responses to the reviewers’ comments you state, “It is a specialist general hospital for the treatment of women's and children's diseases. About 17,000 women give birth in our hospital every year.” This information should be stated in the methods section of the manuscript, too.

---

## [Author Response · Author response to Decision Letter 2]

16 Aug 2022

Dear Editors:

I have revised the manuscript entitled " Risk factors and prognosis of acute lactation mastitis developing into a breast abscess：a retrospective longitudinal study in China." as required by the reviewers. 

We deeply appreciate your consideration of our manuscript, and we look forward to receiving comments from the reviewers. If you have any queries, please don’t hesitate to contact me at the address below.

Thank you and best regards.

Yours sincerely,

Daxue Li and Han Gao

---

## [Editor Report · Decision Letter 3]

19 Aug 2022

Risk factors and prognosis of acute lactation mastitis developing into a breast abscess：a retrospective longitudinal study in China.

PONE-D-21-29131R3

Dear Dr. Gao,

We’re pleased to inform you that your manuscript has been judged scientifically suitable for publication and will be formally accepted for publication once it meets all outstanding technical requirements.

Kind regards,

Forough Mortazavi

Academic Editor

PLOS ONE
---

## [Editor Report · Acceptance letter]

24 Aug 2022

PONE-D-21-29131R3 

Risk factors and prognosis of acute lactation mastitis developing into a breast abscess：a retrospective longitudinal study in China. 

Dear Dr. Gao:

I'm pleased to inform you that your manuscript has been deemed suitable for publication in PLOS ONE. Congratulations! Your manuscript is now with our production department. 

Kind regards, 

on behalf of

Dr. Forough Mortazavi 

Academic Editor

PLOS ONE